**Data Availability Statement:** All relevant data are within the paper an its Supporting Information files.

**Funding:** Unfunded studies.

# Effect of short-term prednisone on beta-cell function in subjects with type 2 diabetes mellitus and healthy subjects

**Monica Shah, May M. Adel, Bettina Tahsin, Yannis Guerra**[ID] **\*, Leon Fogelfeld**

John H. Stroger Jr. Hospital of Cook County, Chicago, Illinois, United States of America

\* yguerra@cookcountyhhs.org

## Abstract

### Objective

For those with type 2 diabetes mellitus (T2DM), impact of short-term high-dose glucocorticoid exposure on beta-cell function is unknown. This study aims to compare the impact on beta-cell function and insulin resistance of prednisone 40 mg between adults with newly diagnosed T2DM and healthy adults.

### Methods

Five adults with T2DM and five healthy adults, all between 18–50 years, were enrolled. T2DM diagnosis was less than one year prior, HbA1c<75 mmol/mol (9.0%), with metformin treatment only. Pre- and post-therapy testing included 75-g oral glucose tolerance, plasma glucose, C-peptide, and insulin. Intervention therapy was prednisone 40mg daily for 3 days.

### Results

Upon therapy completion, HOMA-IR did not increase or differ between groups. Percentile difference for HOMA-%B and insulinogenic index in those with T2DM was significantly lower statistically (50.4% and 69.2% respectively) compared to healthy subjects (19% and 32.2%).

### Conclusions

Contrary to the assumption that insulin resistance is the main driver of glucocorticoid-induced hyperglycemia, results indicate that decreased beta-cell insulin secretion is the more likely cause in those with T2DM. This is evidenced by significant drops in C-peptide AUC and HOMA-%B and increased glucose AUC in T2DM group only. These results may be caused by increased beta-cell fragility along with reduced recovery ability after glucocorticoid exposure. ClinicalTrials.gov NCT03661684.

## Introduction

More than half of patients receiving short-term, high-dose glucocorticoid therapy for conditions like COPD exacerbations, cord compression or post-transplantation experience

**Competing interests:** NO authors have competing interests.

hyperglycemia [1]. This may reflect exacerbation of type 2 diabetes mellitus (T2DM) [2] or can occur in those undiagnosed with diabetes. The hyperglycemia mechanism has been attributed to glucocorticoid-induced insulin resistance and increased hepatic glucose output [3].

Effects of high-dose glucocorticoids on beta-cell function have not been studied extensively. Several studies in healthy men receiving glucocorticoids showed beta-cell function impairment with C-peptide secretion decreased after a short course of high-dose glucocorticoids [4, 5]. No similar studies have been done in people with T2DM. Thus, it is unknown whether and to what extent beta-cell function changes seen in healthy subjects can occur in those with T2DM exposed to short-term high-dose glucocorticoids.

In this study, we aimed to evaluate the effect of a short course of high-dose glucocorticoids on beta-cell function and insulin resistance in adults with new T2DM and assess whether this differs from healthy adults. We chose to use prednisone 40 mg to mirror previous studies in healthy volunteers. [4] This dose also reflects very common short usage of prednisone for intercurrent respiratory infections.

## Materials and methods

### Study subjects

This study used a convenience sample of consecutive five healthy adults and five adults with recent onset T2DM. The study was approved by the institutional review board at John H. Stroger Jr Hospital of Cook County. All participants signed written informed consent before participation.

Eligible subjects with T2DM were between 18 and 50 years of age, diagnosed within 1 year of recruitment, $HbA_{1c} \leq 75$ mmol/mol (9.0%), used only metformin therapy, with BMI of 24–35 kg/m$^2$. Exclusion criteria included estimated glomerular filtration rate (eGFR) <60 mL/min/1.73 m$^2$, glucocorticoid use within six months of recruitment, diabetes therapy other than metformin, shift work, baseline fasting plasma glucose (FPG) $\geq$13.9 mmol/L, or signs or symptoms of infection.

Eligible healthy control subjects were between 18 and 50 years of age, in good physical health per medical history, physical examination, and screening blood tests, were normoglycemic (FPG <5.55 mmol/L), with BMI of 22–28 kg/m$^2$. Exclusion criteria included presence of any chronic disease or any medication use, a first-degree relative with T2DM, history of smoking, glucocorticoids use, shift work, or recent changes in weight or physical activity.

### Study protocol

All subjects (5 subjects with T2DM and 5 healthy subjects) had vitals checked, underwent a physical exam and had comprehensive metabolic panel, lipid panel, and $HbA_{1c}$ drawn. After a 10-hour overnight fast, a 75-g oral glucose tolerance test (OGTT) was done and plasma glucose, C-peptide, and insulin were measured at 0, 30 and 60 minutes. After the OGTT, prednisone 40 mg was administered. Study subjects were given prednisone 40 mg to take at 8am the following day. On the third day, all patients returned to be examined and receive the third dose of prednisone 40 mg. Two hours later, another OGTT and blood samples were collected similarly to the first day.

### Safety protocol

On the first day, all subjects were trained on self-monitoring of capillary glucose and provided glucometers, test strips, lancets, and ketostix. Participants were instructed to check blood glucose levels fasting, pre-prandial (x3 meals), and at bedtime from study day one and up to 2 days

after the second visit. Study coordinator followed-up daily on the blood glucose levels. If blood glucose levels rose above 16.67 mmol/L for more than 2 measurements, subject was withdrawn and blood glucose levels followed by study personnel until it was less than 16.67 mmol/L.

## Outcomes and measures

The primary outcome was beta-cell function and insulin resistance relative changes within and between groups after 3 day exposure of prednisone 40 mg/daily. Beta-cell function measures included OGTT-generated areas under the curve (AUC) of C-peptide levels ($AUC_{C-PEP}$) measured before and after prednisone exposure, as well as the AUC of glucose and insulin levels, an insulinogenic index (IGI) [6], homeostasis model assessment of beta-cell function (HOMA-%B) and homeostasis model assessment of insulin resistance (HOMA-IR) [7].

## Statistical analysis

We based the sample size calculation on the glucocorticoid-induced changes in fasting C-peptide levels found in healthy volunteers in previous studies [4]. We estimated a sample size of 5 participants in each group to provide 80% power to detect a change of 1 unit (nmol/L) between baseline C-peptide and post-glucocorticoid exposure values with a SD of 0.55 and significance level of 0.05.

The AUCs for glucose and C-peptide were calculated using the trapezoidal rule. The IGI, an indicator of early insulin response, was calculated as the ratio of insulin level to glucose response at 0 and 30 minutes ($\Delta I_{30pmol/L}/\Delta G_{30mmol/L}$) [6]. The HOMA-%B and HOMA-IR were calculated as described by Matthews et al using updated online tool [7].

Results were expressed in median and interquartile ranges (IQR) and statistical significance was evaluated using independent samples Mann Whitney U test (intergroup comparisons) and related samples Wilcoxon signed rank test (intragroup comparisons).

## Results

All 10 subjects recruited completed the 3-day prednisone course and the pre- and post-OGTT. Of the five patients with diabetes, three were treated with metformin before and during the study and two were without diabetes medications. Baseline characteristics differed for age, body weight, BMI and blood pressure (systolic and diastolic) between the groups (Table 1). As expected, they also differed in HbA1c.

The AUCs (Glc, C-peptide and insulin), IGI and HOMA-%B and -IR are shown in Table 1. In healthy subjects, there were no statistically significant differences between the pre- and post-prednisone values for any of the variables above. In the subjects with T2DM, the pre- and post-prednisone values were statistically significant for the rise in AUC Glc, decrease in IGI and HOMA-%B. There was also a decrease in the AUC for insulin and C-peptide but it was not significant (both p = 0.08).

The relative percent changes between groups for the AUCs, IGI and HOMA-%B and -IR are shown in Fig 1. In the T2DM group, the increase in AUC Glc was significantly higher (p = 0.016) and the IGI decreased significantly more (69.2% vs 34.3%, p = 0.008). The AUC C-peptide percent decrease more in T2DM group but did not reach significance (p = 0.056). Other variables were not significantly different. The safety glucose monitoring only registered 1 value above 16.67 mmol/L (17.5 mmol/L), in a subject with T2DM, and that subject returned to lower values in the next measurement.

**Table 1. Characteristics and outcomes.**

| Baseline Characteristics | | |
|---|---|---|
| Group | Subjects with T2DM (n = 5) | Healthy Subjects (n = 5) |
| Age (years)* | 46.0 (36.5–48.5) | 29.0 (25.5–31.0) |
| Male gender (n) | 4 (80%) | 2 (40%) |
| Weight (kg)* | 86.9 (74.9–98.0) | 59.1 (56.0–75.7) |
| Height (cm) | 164.0 (158.5–172.7) | 164.0 (159.5–170.9) |
| Body mass index (kg/m$^2$)* | 32.3 (28.9–34.0) | 22.1 (22.0–25.9) |
| Waist Circumference (cm) | 107.0 (100.3–116.5) | 78.5 (75.8–88.0) |
| Systolic BP (mm Hg)* | 128.0 (114.5–134.0) | 109.0 (105.0–115.5) |
| Diastolic BP (mm Hg)* | 87.0 (73.5–94.0) | 71.0 (60.5–77.5) |
| HbA1c (mmol/mol) | 55 (49–58) | 29 (27–32) |
| HbA1c (%)* | 7.2 (6.6–7.5) | 4.8 (4.6–5.1) |
| T2DM Duration (months) | 4.0 (1.0–9.5) | NA |
| eGFR (mL/min/1.73 m$^2$) | 106.0 (91.0–107.0) | 101.0 (93.0–124.5) |
| Outcomes after 3 days of prednisone administration | | | | |
| Group | Subjects with T2DM (n = 5) | | Healthy Subjects (n = 5) | |
| | Baseline | After 3 days of prednisone | Baseline | After 3 days of prednisone |
| FPG (mmol/L)* | 5.9 (5.3–6.4) | 7.0 (6.6–7.2) | 4.8 (4.4–5.1) | 5.2 (4.6–5.9) |
| AUC Glucose (mmol/L/min) | 616.8 (472.1–674.5) | 734.1 (639.8–808.8)[&] | 366.5 (333.5–459.1) | 409.7 (373.9–439.8) |
| AUC C-Peptide (nmol/L/min) | 129.5 (52.3–160.7) | 80.1 (49.1–97.8) | 94.5 (62.2–140.6) | 108.0 (83.8–164.7) |
| AUC Insulin (nmol/L/min) | 31.6 (8.5–49.4) | 13.7 (8.2–21.2) | 23.2 (18.4–39.9) | 17.9 (16.0–42.1) |
| Insulinogenic Index | 66.7 (47.4–132.1) | 24.5 (15.0–37.2)[&] | 258.0 (109.9–452.4) | 164.3 (95.9–438.4) |
| HOMA-IR | 1.57 (0.91–4.12) | 1.35 (1.04–1.59) | 0.65 (0.54–1.1) | 0.71 (0.50–1.03) |
| HOMA-%B | 89.9 (79.4–158.1) | 58.0 (49.5–64.5)[&] | 84.0 (68.5–114.3) | 72.9 (54.7–89.2) |

Data presented as mean ± SD or median (IQR) or absolute value (percentage). FPG, fasting plasma glucose; HbA1c, hemoglobin A1c; eGFR, estimated glomerular filtration rate.

* = p ≤0.05 between groups.

[&] = p≤ 0.05 within groups

## Discussion

In this study, after the administration of prednisone 40 mg/daily for three days, there was decreased beta-cell insulin secretion in subjects with T2DM. This was evidenced by a significant drop in the IGI and HOMA-%B, and near significant decrease in the insulin and C-peptide AUC. The healthy controls did not have significant decreases in beta-cell insulin secretion parameters after prednisone exposures. The relative percent decrease in insulin secretion was also significantly higher in T2DM in comparison to healthy controls. The expected rise in insulin resistance [8] did not occur in either group.

Subjects with T2DM in this study showed vulnerability to short exposure of high-dose glucocorticoids. The increase in glucose AUC was explained by an acute decrease in beta-cell function, mainly related to decreased beta-cell capacity to react to a hyperglycemic stimulus shown by an almost 70% drop in IGI in subjects withT2DM. In healthy subjects on the second day after exposure to one day of prednisone, insulin secretion recovered [4]. Such a recovery can be postulated in our healthy subjects who did not show significant insulin secretion decline after 3 days of prednisone exposure. The acute beta-cell impairment by glucocorticoids in healthy subjects has been shown to be temporary and reversible [4]. The causation may be multifactorial as has been shown in experimental models. In normal mouse islet cells, a

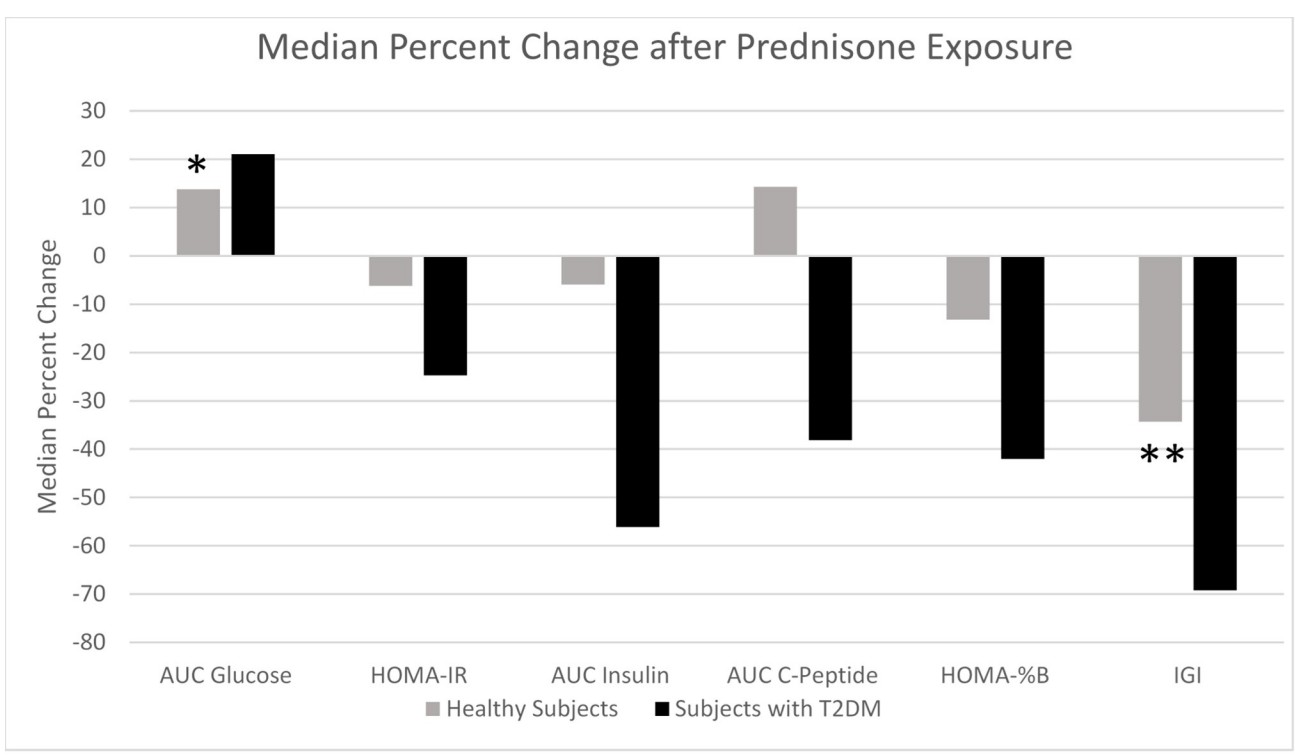

* p<0.05 between groups using Mann Whitney
** p<0.01 between groups using Mann Whitney

| | % change AUC Glucose | % Change HOMA-IR | % change AUC Insulin | % change AUC C-Peptide | % Change HOMA-%B | % Change IGI |
|---|---|---|---|---|---|---|
| Healthy | 13.8 | -6.3 | -6.0 | 14.3 | -13.2 | -34.3 |
| | (-6.7, 15.9) | (-43.7, 109.5) | (-41.5, 58.5) | (-13.0, 82.0) | (-46.3, 20.6) | (-50.3, 86.4) |
| T2DM | 21.0 | -24.8 | -56.1 | -38.1 | -42.0 | -69.2 |
| | (17.3, 38.1) | (-61.2, 29.6) | (-57.2, -7.2) | (-39.3, -4.9) | (-61.5, -29.8) | (-73.7, -64.1) |

Data presented as median (IQR)

**Fig 1. Comparisons of median percent changes of glycemic and beta cell parameters from baseline to after prednisone exposure between the healthy and T2DM groups.**

dexamethasone treatment decreased insulin secretion through a lower efficacy of cytoplasmic calcium on the exocytosis process in the beta cells. [9]. Additionally, glucocorticoids influence the expression of G proteins in different tissues and this may have a role in insulin secretion in the beta cells. [9]. In none of these studies were diabetes models used.

In this study, subjects with T2DM were still affected up to the third day as demonstrated by the drop in IGI and HOMA-%B. Based on our study, one can speculate that subjects with T2DM have an increased beta-cell impairment and reduced ability to recover after glucocorticoid

exposure. Unknown are the duration of beta-cell impairment and whether insulin resistance will increase in multi-day exposure to glucocorticoids, as has been shown in healthy men [10].

This study had several limitations. First, it included a small number of participants in each group, which may explain why compared to previous studies done on healthy patients [3, 4], we did not see a significant difference in their beta-cell function parameters. However, as stated above, the beta-cell emergent response is short-lived in healthy individuals and recovered at the third day. Second, the two groups had an age mismatch. We had limited ability to find younger patients with T2DM who did not have the exclusion criteria of BMI >35.0 kg/m$^2$, likely due to the association of obesity to the pathophysiology of T2DM, especially in younger people. We consider this a reasonable tradeoff, as higher BMI values may affect the insulin resistance value more than older ages, obscuring a possible effect of the glucocorticoids.

In conclusion, these results show that a short course of high-dose glucocorticoids in those with T2DM leads to hyperglycemia due to a significant decrease in insulin secretion. Further studies with larger cohorts and longer exposure to glucocorticoids are needed to clarify what pathophysiological mechanisms operate over time in these patients.

## Supporting information

**S1 File.**
(DOC)

**S2 File.**
(DOCX)

**S3 File.**
(DOC)

**S4 File.**
(DOC)

**S1 Data.**
(XLSX)

## Acknowledgments

We want to give thanks to Tony Wicheanvonago for his expertise of lab sample handling.

## Author Contributions

**Conceptualization:** Monica Shah, Yannis Guerra, Leon Fogelfeld.

**Data curation:** Monica Shah, May M. Adel, Yannis Guerra, Leon Fogelfeld.

**Formal analysis:** Monica Shah, Bettina Tahsin, Yannis Guerra, Leon Fogelfeld.

**Investigation:** Monica Shah, May M. Adel, Yannis Guerra, Leon Fogelfeld.

**Methodology:** Monica Shah, May M. Adel, Bettina Tahsin, Yannis Guerra.

**Project administration:** May M. Adel, Yannis Guerra, Leon Fogelfeld.

**Resources:** Bettina Tahsin, Yannis Guerra, Leon Fogelfeld.

**Supervision:** Yannis Guerra, Leon Fogelfeld.

**Validation:** Bettina Tahsin, Yannis Guerra, Leon Fogelfeld.

**Writing – original draft:** Monica Shah, Yannis Guerra, Leon Fogelfeld.

**Writing – review & editing:** Monica Shah, May M. Adel, Bettina Tahsin, Yannis Guerra, Leon Fogelfeld.

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
