## [Decision Letter · Decision Letter 0]

31 Dec 2019

PONE-D-19-24296

Effect of short-term prednisone on beta-cell function in subjects with type 2 diabetes mellitus and healthy subjects

PLOS ONE

Dear Dr Guerra,

Thank you for submitting your manuscript to PLOS ONE. After careful consideration, we feel that it has merit but does not fully meet PLOS ONE’s publication criteria as it currently stands. Therefore, we invite you to submit a revised version of the manuscript that addresses the points raised during the review process.

We would appreciate receiving your revised manuscript by Feb 14 2020 11:59PM. To enhance the reproducibility of your results, we recommend that if applicable you deposit your laboratory protocols in protocols.io, where a protocol can be assigned its own identifier (DOI) such that it can be cited independently in the future. For instructions see: http://journals.plos.org/plosone/s/submission-guidelines#loc-laboratory-protocols

We look forward to receiving your revised manuscript.

Kind regards,

Zhiming Zhu

Academic Editor

PLOS ONE

Journal Requirements:

Reviewers' comments:

Reviewer's Responses to Questions

**Comments to the Author**

1. Is the manuscript technically sound, and do the data support the conclusions?

Reviewer #1: No

Reviewer #2: Yes

Reviewer #3: Yes

2. Has the statistical analysis been performed appropriately and rigorously? 

Reviewer #1: No

Reviewer #2: Yes

Reviewer #3: Yes

3. Have the authors made all data underlying the findings in their manuscript fully available?

Reviewer #1: Yes

Reviewer #2: Yes

Reviewer #3: Yes

4. Is the manuscript presented in an intelligible fashion and written in standard English?

Reviewer #1: Yes

Reviewer #2: Yes

Reviewer #3: Yes

5. Review Comments to the Author

Reviewer #1: Although the authors addressed a clinical issue about the effect of prednisone on beta-cell function between T2DM patients and healthy people, the sample size was too small to support the conclusion, and the mismatch of age and gender lead to large sample bias. To the reviewer's opinion, more rigorous experiments and data were needed to confirm the conclusions.

Reviewer #2: Shah and colleagues drew a preliminary but interesting study concluding high-dose prednisone resulting in suppressed insulin secretion rather than increasing insulin resistance. The conclusion seems result-based, but some concerns should be noted before its publication.

1. How did the author confirm the diagnosis of type 2 diabetes? As this is a very small sample trial, the representation of the study is exclusively important (authors may wish to add differentiating information like auto-antibodies, baseline 8 am cortisol and other MODY phenotypes).

2. The therapeutic regimens need to be provided including a daily dose of metformin. If there is any dose change of metformin during the trial, it should be reported.

3. Are there any other safety issues reported in the trial, such as hypertension, insomnia, etc?

4. The reduction of AUC C-peptide and AUC Insulin is not totally in line. Please explain.

5. The authors may wish to provide some clinical interpretations in steroid usage, especially the clinical context of a three-day prescription of 40mg/d prednisone.

6. The authors may also wish to discuss the underlying pathophysiological mechanism of the study result in more detail.

7. This trial does not include a dose gradient of prednisone, which is also a limitation needing stated.

8. I think "emergent response of beta-cell function" rather than "short-short effects" may more fit the study context.

9. Please note some typos like OGGT in the manuscript.

Reviewer #3: In their manuscript entitled “Effect of short-term prednisone on beta-cell function in subjects with type 2 diabetes mellitus and healthy subjects," Shah and colleagues report that that a short course of high-dose glucocorticoids in those with T2DM leads to a significant decrease in insulin secretion. However, the healthy controls did not have significant decreases in beta-cell insulin secretion parameters after prednisone exposures. The authors believe that subjects with T2DM have an increased beta-cell impairment and reduced ability to recover after glucocorticoid exposure.

Overall, the authors report a very interesting results; however, a few issues do require attention in my opinion.

1. In fact, lots of studies have demonstrated that glucocorticoids treatment may lead to insulin resistance, increased hepatic gluconeogenesis and hyperglycemia in animals and human. Of note, these effects of glucocorticoids are dependent on long term treatment (for example, 2 weeks or 3 weeks). However, in the present study, human subjects were treated with glucocorticoid for 3 days. Thus, the authors just observed a short effects of glucocorticoid treatment. These data are interesting. However, when the authors make conclusion, they should be more objective and fair. In my opinion, they should conclude that short-term glucocorticoid treatment may impair insulin secretion and beta cell in patients with type 2 diabetes. They can not deny the long term effects of glucocorticoids on insulin resistance and hepatic gluconeogenesis.

2. The mechanism of this study is completely unclear, and the mechanism of how glucocorticoid affects the decrease of insulin secretion by beta cells in diabetic patients is unclear. The authors should discuss this issue, based on the previous literatures.

3. The number of participants is not enough. The author also mentioned this point and acknowledged this weakness.

4. What were the results in animals? Has the authors performed similar experiments in mice?

6. PLOS authors have the option to publish the peer review history of their article (what does this mean?). If published, this will include your full peer review and any attached files.

Reviewer #1: No

Reviewer #2: No

Reviewer #3: No

---

## [Author Response · Author response to Decision Letter 0]

13 Feb 2020

5. Review Comments to the Author

Reviewer #1: Although the authors addressed a clinical issue about the effect of prednisone on beta-cell function between T2DM patients and healthy people, the sample size was too small to support the conclusion, and the mismatch of age and gender lead to large sample bias. To the reviewer's opinion, more rigorous experiments and data were needed to confirm the conclusions.

We agree with the concerns of Reviewer #1. The size used was based on our sample calculation for the limited outcomes measured. Despite these concerns there were significant differences between the two groups. To avoid type 1 error, there is a need to confirm the study findings with bigger and better matched groups. We indicated this in the limitations (page 8, line 159-161) and in the conclusion (page 9, line 172). 

Reviewer #2: Shah and colleagues drew a preliminary but interesting study concluding high-dose prednisone resulting in suppressed insulin secretion rather than increasing insulin resistance. The conclusion seems result-based, but some concerns should be noted before its publication.

1. How did the author confirm the diagnosis of type 2 diabetes? As this is a very small sample trial, the representation of the study is exclusively important (authors may wish to add differentiating information like auto-antibodies, baseline 8 am cortisol and other MODY phenotypes).

We confirmed the diagnosis of type 2 diabetes using the standard criteria of A1c above 6.5% (present in 5/5 cases and 0/5 controls). In addition, our patients underwent OGTT, which showed criteria for diagnosis of DM2 in 4/5 cases and 0/5 controls. All patients had normal C-peptide production levels at baseline time for OGTT, which would rule out significant autoimmune diabetes as the cause of hyperglycemia. MODY phenotypes were not likely since those with diabetes were relatively well-controlled and none were treated with sulfonylurea medications. 

2. The therapeutic regimens need to be provided including a daily dose of metformin. If there is any dose change of metformin during the trial, it should be reported.

Of the five patients with diabetes, three were treated with metformin before and during the study and two were without diabetes medications (added to pages 5-6, lines 105-107); there was no change to any doses of medications. Patients were informed of the importance of continuing all their medications during the time of the prednisone administration. 

 

3. Are there any other safety issues reported in the trial, such as hypertension, insomnia, etc?

No other safety issues were reported per patient after interview with research assistant on day of 3rd dose administration. 

4. The reduction of AUC C-peptide and AUC Insulin is not totally in line. Please explain.

The reduction of AUC of both C-peptide and insulin were in the same direction for the group of subjects with diabetes. For the healthy subjects, the explanation for discordant direction of the insulin and C-peptide may lie in the outlier values of two patients in whom both the insulin and C-peptide went up but only minimally.

5. The authors may wish to provide some clinical interpretations in steroid usage, especially the clinical context of a three-day prescription of 40mg/d prednisone.

The dose selected was similar to that used in study of healthy volunteers. This is also the dose recommended for short courses of steroids in case of COPD or asthma exacerbation. This explanation is included in the introduction (page 3, lines 44-45).

6. The authors may also wish to discuss the underlying pathophysiological mechanism of the study result in more detail.

Discussion of the mechanisms causing the beta cell impairment was amplified (page 8, lines 145-149). 

7. This trial does not include a dose gradient of prednisone, which is also a limitation needing stated.

The few data available show that dose ranges from 30-75 mg/day do not seem to cause different effects in patients without diabetes mellitus. See page 3, lines 42-45 for rationale of dosing.

8. I think "emergent response of beta-cell function" rather than "short-short effects" may more fit the study context.

Wording was changed on page 9, line 162 to incorporate this suggestion.

9. Please note some typos like OGGT in the manuscript.

Corrections made. 

 

Reviewer #3: In their manuscript entitled “Effect of short-term prednisone on beta-cell function in subjects with type 2 diabetes mellitus and healthy subjects," Shah and colleagues report that that a short course of high-dose glucocorticoids in those with T2DM leads to a significant decrease in insulin secretion. However, the healthy controls did not have significant decreases in beta-cell insulin secretion parameters after prednisone exposures. The authors believe that subjects with T2DM have an increased beta-cell impairment and reduced ability to recover after glucocorticoid exposure.

Overall, the authors report very interesting results; however, a few issues do require attention in my opinion.

1. In fact, many studies have demonstrated that glucocorticoids treatment may lead to insulin resistance, increased hepatic gluconeogenesis and hyperglycemia in animals and human. Of note, these effects of glucocorticoids are dependent on long term treatment (for example, 2 weeks or 3 weeks). However, in the present study, human subjects were treated with glucocorticoid for 3 days. Thus, the authors just observed short effects of glucocorticoid treatment. These data are interesting. However, when the authors make conclusion, they should be more objective and fair. In my opinion, they should conclude that short-term glucocorticoid treatment may impair insulin secretion and beta cell in patients with type 2 diabetes. They cannot deny the long-term effects of glucocorticoids on insulin resistance and hepatic gluconeogenesis.

We agree with reviewer, that the long term effect of the glucocorticoids in healthy volunteers has been shown multiple times. Our study is the first to evaluate the response in patients with type 2 diabetes, although only for the initial 72 hours of administration. We agree that studies still need to be done for effects over the long term. Edits have been made on page 9, lines 170-172 to reflect this.

2. The mechanism of this study is completely unclear, and the mechanism of how glucocorticoid affects the decrease of insulin secretion by beta cells in diabetic patients is unclear. The authors should discuss this issue, based on the previous literatures

Discussion of the mechanisms causing the beta cell impairment was amplified (page 8, lines 145-149). 

3. The number of participants is not enough. The author also mentioned this point and acknowledged this weakness.

The size used was based on our sample calculation for the limited outcomes measured. Despite these concerns there were significant differences between the two groups. To avoid type 1 error, there is a need to confirm the study findings with bigger and better matched groups. We indicated this in the limitations (page 8, line 159-161) and in the conclusion (page 9, line 172).

4. What were the results in animals? Has the authors performed similar experiments in mice?

We are exclusively a clinical research site. Data on animals has been done before (Lambillote et al). We had hoped to show the effect of these drugs on a population that has not been studied directly before (humans with type 2 diabetes).

---

## [Decision Letter · Decision Letter 1]

19 Mar 2020

Effect of short-term prednisone on beta-cell function in subjects with type 2 diabetes mellitus and healthy subjects

PONE-D-19-24296R1

Dear Dr. Guerra,

We are pleased to inform you that your manuscript has been judged scientifically suitable for publication and will be formally accepted for publication once it complies with all outstanding technical requirements.

With kind regards,

Zhiming Zhu

Academic Editor

PLOS ONE

Additional Editor Comments (optional):

Reviewers' comments:

Reviewer's Responses to Questions

**Comments to the Author**

1. If the authors have adequately addressed your comments raised in a previous round of review and you feel that this manuscript is now acceptable for publication, you may indicate that here to bypass the “Comments to the Author” section, enter your conflict of interest statement in the “Confidential to Editor” section, and submit your "Accept" recommendation.

Reviewer #1: All comments have been addressed

Reviewer #2: All comments have been addressed

Reviewer #3: All comments have been addressed

2. Is the manuscript technically sound, and do the data support the conclusions?

Reviewer #1: Partly

Reviewer #2: No

Reviewer #3: Yes

3. Has the statistical analysis been performed appropriately and rigorously? 

Reviewer #1: Yes

Reviewer #2: Yes

Reviewer #3: Yes

4. Have the authors made all data underlying the findings in their manuscript fully available?

Reviewer #1: Yes

Reviewer #2: Yes

Reviewer #3: Yes

5. Is the manuscript presented in an intelligible fashion and written in standard English?

Reviewer #1: No

Reviewer #2: Yes

Reviewer #3: Yes

6. Review Comments to the Author

Reviewer #1: (No Response)

Reviewer #2: Thanks for the careful revision from the authors. However, I would doubt the credibility of the study results as the authors suggested that the AUC of CP and insulin may lie when there the sample size is too small. Meanwhile, I saw only diabetes but not type 2 diabetes in this study. Both issues may lead to the study results, which are different from our clinical impression.

I would suggest the authors to add sample size based on the current results and rigorously exclude potential MODY and LADA patients.

Reviewer #3: (No Response)

7. PLOS authors have the option to publish the peer review history of their article (what does this mean?). If published, this will include your full peer review and any attached files.

Reviewer #1: No

Reviewer #2: No

Reviewer #3: No

---

## [Editor Report · Acceptance letter]

9 Apr 2020

PONE-D-19-24296R1 

Effect of short-term prednisone on beta-cell function in subjects with type 2 diabetes mellitus and healthy subjects 

Dear Dr. Guerra:

I am pleased to inform you that your manuscript has been deemed suitable for publication in PLOS ONE. Congratulations! Your manuscript is now with our production department. 

With kind regards,

on behalf of

Dr. Zhiming Zhu 

Academic Editor

PLOS ONE